# Cognitive Impairment in Nonagenarians: Potential Metabolic Mechanisms Revealed by the Synergy of In Silico Gene Expression Modeling and Pathway Enrichment Analysis

**DOI:** 10.3390/ijms25063344

**Published:** 2024-03-15

**Authors:** Aleksandra Mamchur, Elena Zelenova, Irina Dzhumaniiazova, Veronika Erema, Daria Kashtanova, Mikhail Ivanov, Maria Bruttan, Mariia Gusakova, Mikhail Terekhov, Vladimir Yudin, Antonina Rumyantseva, Lorena Matkava, Irina Strazhesko, Ruslan Isaev, Anna Kruglikova, Lilit Maytesyan, Irina Tarasova, Olga Beloshevskaya, Elen Mkhitaryan, Sergey Kraevoy, Olga Tkacheva, Sergey Yudin

**Affiliations:** 1Centre for Strategic Planning and Management of Biomedical Health Risks, Federal Medical Biological Agency, bld.10/1, Pogodinskaya Str., Moscow 119121, Russia; amamchur@cspfmba.ru (A.M.);; 2Russian Clinical Research Center for Gerontology, Pirogov Russian National Research Medical University of the Ministry of Healthcare of the Russian Federation, bld. 16, 1st Leonova Street, Moscow 129226, Russia

**Keywords:** cognitive impairment, dementia, MMSE, long-living adults, TWAS, longevity, functional analysis

## Abstract

Previous studies examining the molecular and genetic basis of cognitive impairment, particularly in cohorts of long-living adults, have mainly focused on associations at the genome or transcriptome level. Dozens of significant dementia-associated genes have been identified, including APOE, APOC1, and TOMM40. However, most of these studies did not consider the intergenic interactions and functional gene modules involved in cognitive function, nor did they assess the metabolic changes in individual brain regions. By combining functional analysis with a transcriptome-wide association study, we aimed to address this gap and examine metabolic pathways in different areas of the brain of older adults. The findings from our previous genome-wide association study in 1155 older adults, 179 of whom had cognitive impairment, were used as input for the PrediXcan gene prediction algorithm. Based on the predicted changes in gene expression levels, we conducted a transcriptome-wide association study and functional analysis using the KEGG and HALLMARK databases. For a subsample of long-living adults, we used logistic regression to examine the associations between blood biochemical markers and cognitive impairment. The functional analysis revealed a significant association between cognitive impairment and the expression of NADH oxidoreductase in the cerebral cortex. Significant associations were also detected between cognitive impairment and signaling pathways involved in peroxisome function, apoptosis, and the degradation of lysine and glycan in other brain regions. Our approach combined the strengths of a transcriptome-wide association study with the advantages of functional analysis. It demonstrated that apoptosis and oxidative stress play important roles in cognitive impairment.

## 1. Introduction

Genetic predisposition plays a significant role in the development of cognitive impairment. Many genome-wide association studies (GWASs) have explored how genes and their single nucleotide variants (SNVs) affect the likelihood and severity of this condition. The APOE gene and its variants have been linked to different types of dementia, including late-onset Alzheimer’s disease [1], vascular dementia [2], frontotemporal dementia [3], and Lewy body dementia (LBD) [4]. In our previous GWAS, we discovered eight SNVs on chromosomes 1, 4, and 19 and examined the effects of the missense mutation, rs429358, on the APOE protein’s structure and function, specifically its lipid-binding domain [5]. Notably, significant SNVs are not limited to missense mutations. Most of them are located in non-coding regions. Hence, these SNVs do not directly affect protein structure or function but exert regulatory effects. Their effects on the phenotype should be further examined in studies other than GWASs.

Transcriptome-wide association studies (TWASs) enable the measuring of tissue-specific changes in individual gene expression levels and provide additional benefits for assessing SNV effects. Sampling a living brain with cognitive impairment poses a great challenge, but in silico techniques help circumvent this limitation. Methods like PrediXcan [6], Sherlock [7], and ENLOC [8] use GWAS findings to predict gene expression levels. They generate coefficients reflecting the combined effect of SNVs on tissue-specific gene expression levels. TWASs of Alzheimer’s disease have discovered dozens of “causal” tissue-specific genes, such as *ACE*, *APOE*, *APOC1*, *FAM241A*, *SAPCD1*, *FAM111A*, and *TOMM40* [9]. However, these studies do not specify the type of tissue or the extent of changes in gene expression levels, nor do they consider intergenic interactions or functional gene modules involved in cognitive dysfunction. Therefore, no well-founded conclusions about the biology of the respective phenotypes can be drawn from these results alone. Functional analysis is used to determine how SNVs affect gene expression dynamics and form a specific phenotype. It has been used as a standalone technique to find associations between Alzheimer’s disease and functional signaling pathways from the Gene Ontology Annotation (GOA) database and the Kyoto Encyclopedia of Genes and Genomes (KEGG) database [10]. In GOA, Alzheimer’s disease has been linked to plasma lipoprotein assemblies, reverse cholesterol transport (RCT), catabolic processing of amyloid precursor protein (APP), and activation of the immune response [10]. Chouliaras et al. performed a gene ontology (GO) analysis and KEGG pathway enrichment analysis that revealed a number of pathways associated with cognitive dysfunction and dementia, such as “main axon” and “β-amyloid binding” in GOA and “glutamatergic synapse” and “Alzheimer’s disease” in KEGG. These findings suggest that the genes involved in these pathways may contribute to cognitive impairment [11]. However, the above studies measured gene expression levels in whole blood and not in brain tissue, which limits the applicability of their findings.

In this study, we used both a TWAS and functional analysis to measure the changes in brain metabolic pathways (Figure 1). This dual approach helped address the gaps in our understanding of the underlying molecular mechanisms of cognitive impairment in long-living adults.

## 2. Results

### 2.1. The Cohort

The study included 179 participants with cognitive impairment and 976 cognitively healthy participants (*n* = 1155) (Figure 2). A total of 74.4% of the cohort were women. Overall, 91.2% of men and 82.2% of women were cognitively healthy (*p*-value = 0.00013).

Table 1 shows associations between the blood biochemical markers and cognitive status. A 1 µ/L increase in insulin resulted in a 6% decrease in the incidence of cognitive impairment (OR = 0.94; *p*-value = 0.009). A one-unit increase in Apolipoprotein A (OR = 0.97; *p*-value = 1.91 × 10^−14^), HDL (OR = 0.22; *p*-value = 5.76 × 10^−7^), IGF-1 (OR = 0.99, *p*-value = 0.0008), and 25(OH)D (OR = 0.89, *p*-value = 7.74 × 10^−7^) also showed protective effects and decreased the incidence of cognitive impairment by 3%, 78%, 1%, and 11%, respectively.

Notably, this study sample included long-living adults of a very advanced age (90+ years). There are generally more women in this age group, as reflected in our cohort. However, men over 90 years of age are cognitively healthier than their female counterparts. Blood plasma biomarkers such as insulin, apolipoprotein A, HDL, insulin-like growth factor, and vitamin D were associated with cognitive impairment.

### 2.2. TWAS

Figure 3 shows the TWAS results for the anterior cingulate cortex. The most significant association was found between cognitive impairment and lower expression levels of *LRRC25* (z-score = −4.118; *p*-value = 3.83 × 10^−5^). The significance of the associations between cognitive impairment and increased expression levels of *DLX6* and *PRB2* was marginally below the threshold (z-score = 3.582; *p*-value = 3.41 × 10^−4^
*PRB2*; z-score = 3.699; *p*-value = 2.17 × 10^−4^, respectively). Appendix A presents the most significant differently expressed genes in all brain parts.

Figure 4 shows changes in the predicted expression levels of the selected genes in various brain tissues. In cognitively healthy participants, the expression levels of most genes did not change in any brain tissue. However, the expression levels of *ADSL*, *CTSF*, *LPIN1*, *BCS1L*, and *DISP1* mostly increased. In addition, adjacent areas of the central nervous system formed separate clusters. For example, the spinal cord tissues at the level of C1 and the brain tissues were in different clusters.

Based on the data presented in Figure 4, we were able to assess the quality of the results obtained by comparing changes in the expression levels of the same genes across different brain tissues. For this purpose, in addition to gene clustering, we also carried out tissue clustering (Figure 4, *Y* axis). Given that the spinal cord tissues are brain tissues, the patterns of gene expression changes in the spinal cord differed from those in all other tissues. Tissues in the cerebellum formed a separate cluster, despite exhibiting slight differences from each other, which is inevitable when using bulk RNA-seq data from GTEx. This finding partially validates our TWAS predictions of changes in gene expression levels that occur in cognitive impairment.

Based on the TWAS, the LRRC25 gene was the only gene that showed a statistically significant association with cognitive impairment. For an in-depth examination of the results obtained, we carried out functional analysis.

### 2.3. Functional Analysis

Table 2 presents the results of the functional analysis.

We assessed the expression levels of genes, the products of which are involved in significant functional pathways. In the lysine degradation pathway, the levels of most enzymes increased (Figure 5). The levels of enzymes that catalyze glycan degradation mostly decreased (Figure 6).

In participants with cognitive impairment, the expression levels of KEGG:M00146’s *NDUFA2*, *NDUFA3*, *NDUFA4*, *NDUFA6*, *NDUFA7*, *NDUFA10*, and *NDUFA12* increased, but the expression levels of *NDUFA5*, *NDUFA4L2*, and *NDUFAB1* marginally decreased (Figure 7). The pathway for the synthesis of the NADH dehydrogenase 1-alpha subcomplex was generally more active in the cerebral cortex and particularly in the frontal cortex.

The expression levels of genes in the hsa04215 functional pathway (apoptosis) changed to varying degrees (Figure 8). The expression levels of *BSL2L1*, *BSL2L11*, and *MAP10* marginally decreased, while those of *BAK1*, *BCL2*, *CASP3*, *BOK*, *CASP8*, *MAPK8*, *BIRC6*, and *BIRC7* increased. The expression levels of *APAF1*, *BBC3*, *BIRC5*, and *MAPK9* did not change.

The expression levels of genes in HALLMARK_PEROXISOME varied. The expression levels of *SOD1* and most ATP carriers decreased. The expression levels of the *PEX* genes increased (Appendix A).

Thus, the functional analysis of the TWAS results revealed five metabolic pathways that showed the most significant associations with cognitive impairment. Two of these pathways (HALLMARK_PEROXISOME and NADH-ubiquinone oxidoreductase) are associated with cellular oxidative processes during energy metabolism, including oxidative stress. The lysine and glycan degradation pathways are also associated with catabolism. Apoptosis was the fifth most significant process associated with cognitive impairment.

## 3. Discussion

Consistent with the well-known male–female health–survival paradox, there were significantly more long-living women in our study. However, more men were cognitively healthy (43.6% vs. 36.4%; *p*-value = 0.00013). This finding suggests that men may be less susceptible to age-related cognitive impairment than women. 

LRRC25 was the most significant gene in many areas of the brain, including the anterior cingulate cortex, cerebellum, cerebral cortex, hypothalamus, spinal cord at the level of C1, and the substantia nigra (Figure 3). It is located on chromosome 19 and encodes a leucine-rich repeat protein (LRR). This gene has been shown to inhibit NF-κB signaling [12]. Elevated levels of NF-κB and the resulting increased signaling have been observed in patients with Alzheimer’s disease [13]. The expression of LRRC25 in microglial cells is reduced in Alzheimer’s disease patients carrying disease-associated SNVs [14]. We obtained similar results. Thus, reduced LRRC25 expression may enhance NF-κB signaling, induce apoptosis, and cause pathological conditions such as inflammation. Moreover, a recent study by Feng et al. demonstrated an association between LRRC25 and autophagy activation [12]. Lower expression levels of LRRC25 may inhibit autophagy, hindering cells from eliminating misfolded proteins and aggregates, such as amyloids typical of Alzheimer’s disease. Therefore, this pathway could be a valuable therapeutic target. This assumption, however, requires experimental verification. Increased LRRC25 expression in the cell model of cerebral ischemia has been shown to have a specific effect on this disease that is known to contribute to cognitive deficits [15]. In our study, cognitive impairment was probably associated with neurodegenerative processes rather than vascular disorders. Interestingly, *APOE* and *APOC1*, which showed genome-wide significance in the GWAS [5], were less significant in the TWAS.

A number of KEGG and HALLMARK pathways were highly significant for the spinal cord at the level of C1, the cerebellum, and the nucleus accumbens (Table 2). One of these functional pathways is involved in apoptosis, which may underlie the decrease in LRRC25 expression levels described above (Figure 8). Previous studies have shown that apoptosis is often associated with neurodegenerative diseases such as Alzheimer’s [16]. Both the buildup of amyloid plaque and neurofibrillary tangles and severe oxidative stress can cause neuroapoptosis [16]. In Alzheimer’s disease, caspase-8 and caspase-9 become colonized in the brain, resulting in mitochondrial dysfunction [16]. This correlates with the elevated CASP8 levels in our study (Figure 8). The predicted expression level of BAK1 is higher in participants with cognitive impairment. A BAK1-encoded protein promotes the opening of the mitochondrial voltage-dependent anion channel. This results in loss of mitochondrial membrane potential, cytochrome C release, production of reactive oxygen species (ROS), and activated apoptosis. Increased BAK1 expression has been linked to auditory dysfunction in patients with Alzheimer’s disease [16,17]. Gene products involved in apoptosis are potential therapeutic targets for this condition [18,19]. However, necroptosis and other forms of cell death have also been explored as potential targets [19].

The HALLMARK_PEROXISOME pathway includes genes involved in peroxisome assembly and function. This pathway was highly significant, indicating an association between the production of reactive oxygen species (ROS) and CI. Peroxisomes contain many ROS-producing oxidases; however, they also contain antioxidant enzymes [20]. We found decreased SOD1 expression levels, indicating an impaired protective mechanism against oxidative stress in peroxisomes (Appendix A). In murine models, peroxisome dysfunction has been implicated in neurodegenerative disorders, including Alzheimer’s disease. Peroxisomes are the main target of amyloid plaques and ROS-mediated damage [21].

ROS and mitochondrial dysfunction have been linked to dementia [22]. Notably, the pathway for increased NADH-ubiquinone oxidoreductase (mitochondrial complex I) expression was significant in the cerebral cortex and, particularly, in the frontal cortex (Figure 7).

These findings may indicate that this complex is hyperactive in patients with dementia, resulting in increased ROS production and oxidative stress. Complex 1 has been a potential therapeutic target for metabolic dysfunction. For example, metformin is an antidiabetic that inhibits NADH oxidoreductase [23]. A meta-analysis has shown that this drug can also have a neuroprotective effect in older adults [24]. Therefore, complex 1 is also considered a therapeutic target for dementia, including Alzheimer’s disease [25]. Many hormonal and metabolic factors affect oxidative stress and, thus, cognitive status. For instance, androgens can inhibit oxidative stress [26], which may account for higher MMSE scores among older men compared to older women.

The protective effects of elevated 25-OH vitamin D (calcidiol) levels may result from ROS-related processes that underlie dementia. Calcidiol was reported to have antioxidant properties [27]. However, clinical trials of 25-OH vitamin D supplementation have not conclusively confirmed its antioxidative effect [28]. We found that a 1 ng/mL increase in 25-OH vitamin D reduced the risk of cognitive impairment by 11% (Table 1). 

In our study, increased insulin levels were negatively correlated with cognitive impairment. High circulating insulin levels and insulin resistance are known to contribute to cognitive impairment in the old and oldest-old [29]. Substantial oxidative stress has been observed in the brains of older people under 90 years of age with diabetes and elevated insulin levels, which may underlie the link between diabetes and dementia [30]. This is not the case over the age of 90, when a negative correlation between insulin and cognitive impairment arises, suggesting its protective property in this age group [31]. The link between insulin levels, oxidative stress, and dementia requires further research. In people aged 90 and older, cognitive impairment is correlated only with HDL, whereas correlations with total cholesterol, triglycerides, and LDL do not reach the significance threshold [32]. Apolipoprotein A is a major structural protein in high-density lipoproteins. Therefore, HDL and Apo-A have similar effects on cognitive impairment (Table 1). Studies in animal models have shown that ApoA1 knockdown increases inflammation and vascular amyloid-beta deposition in the brain. Our results and the published literature suggest that Apolipoprotein A has a protective effect against various diseases, including neurodegenerative disorders.

IGF-1 levels tend to decrease with age and negatively correlate with cognitive impairment. IGF-1 is believed to affect the morphology and function of synapses as well as the excitability of neurons via signaling pathways [33]. Moreover, IGF-1 has been shown to have a neuroprotective effect. For example, it promoted the survival of primary cerebellar neurons in rats and hypothalamic cells in immortalized GT1–7 rats after hydrogen peroxide (H_2_O_2_)-induced oxidative stress [34]. This effect is mediated by nuclear factor-κB (NF-κB) [34]. In our study, high IGF-1 levels significantly reduced the risk of cognitive impairment. Notably, IGF-1 has a critical effect on apoptotic processes [35].

There was a significant association between cognitive impairment and another glycan degradation pathway, hsa00511. Yu et al. proposed using N-glycans as biomarkers for the early diagnosis of Alzheimer’s disease [36]. Glycan degradation is known to promote macroautophagy [37]. Most genes associated with this process (Figure 6), such as *FUCA1*, *NEU1*, *MAN2C1*, *GLB1*, *HEXA*, *AGA*, and *MAN2B*2, had lower expression levels in the TWAS. Consistent with previously published data [38], this finding may indicate impaired autophagy in the analyzed brain tissues of patients with cognitive impairment.

We observed increased expression levels of almost all enzymes involved in the L-lysine degradation pathways (Figure 5). L-lysine is an essential amino acid. Its impaired catabolism can cause disorders such as Glutaric aciduria type 1 (GA1) [39,40], leading to damaged subcortical brain structures. Interestingly, an increased rate and intensity of amino acid metabolism may be a compensatory response to glucose deficiency. This is often observed in patients with Alzheimer’s disease [41]. Thus, the predicted higher expression levels of enzymes involved in amino acid catabolism may constitute a compensatory mechanism for supplying energy to neurons. This suggestion requires further investigation and experimental verification.

The detected associations between cognitive impairment and the expression levels of certain genes were statistically significant. However, to establish the causality, if any, underlying these associations, the presented findings need to be tested in future studies. The presented findings may facilitate future experiments on the molecular and genetic bases of cognitive impairment.

## 4. Materials and Methods

### 4.1. Participants and Examination Procedures

This non-interventional, cross-sectional study analyzed a sample of 1155 long-living individuals aged 90 years and older who were recruited between 2019 and 2021 with the assistance of social and geriatric facilities in Moscow and the Moscow region. The participants completed geriatric scales and questionnaires with the assistance of a physician and registered nurse at their places of residence and had their biomaterials collected (including whole blood for the GWAS). Blood samples were analyzed for key biochemical markers: LDL, HDL, cholesterol, glucose, glycosylated hemoglobin, apolipoproteins Apo-A1 and Apo-B, insulin-like growth factor 1 (IGF-1), cortisol, 25(OH)D, and insulin using the enzymatic method, the Friedewald formula (FF), photometry, capillary electrophoresis, immunoturbidimetry, and chemiluminescence detection, respectively. A more detailed description of the study design is provided in our previous study [42].

Cognitive status was assessed using the Mini-Mental State Examination (MMSE), with a score of ≤9 indicating cognitive impairment (encoded as 1) and a score of >24 indicating cognitive health (encoded as 0) [43,44]. Cognitive impairment was encoded as 1, and cognitive health was encoded as 0. Due to the many confounders typical of the examined age range (sensory deficit, easy fatigability, etc.) and hard-to-interpret cutoff values, this binary approach was applied in all further analyses, including GWAS, TWAS, the functional analysis, and the analysis of associations between the biomarkers and cognitive impairment.

### 4.2. Statistical Analysis

The associations between the blood biochemical markers and cognitive impairment were assessed using logistic regression in Statsmodels [45] (Python 3.9.12) (Table 1). Body mass index (BMI), gender, and age were used as covariates, and Bonferroni adjustment was applied for multiple testing correction. Each logistic regression coefficient was exponentiated to calculate odds ratios (ORs). The significance threshold was set at ≤ 0.05.

The clustermap function in Seaborn 0.12.2 was used for data clustering and visualization (Figure 4) [46]. The images show genes that are expressed in all brain tissues. These genes have been associated with cognitive impairment-related disorders, such as Alzheimer’s disease, autism, and congenital brain malformations, and have “reviewed by expert panel” status in ClinGen [47].

### 4.3. Transcriptome-Wide Association Study (TWAS)

The GWAS results (SNVs, logistic regression coefficients, and *p*-values) [5] and eQTL data from GTEx [48] were used as input. Gene expression levels were predicted using PrediXcan (https://github.com/hakyimlab/MetaXcan (accessed on 7 March 2023)), an algorithm that computes gene expression changes, or z-scores, in a specific tissue type.

The algorithm predicted gene expression levels in 49 tissue types. The following brain areas with available tissue-specific gene expression levels in GTEx were selected for further analysis: the spinal cord at the level of C1 (first cervical vertebra), the cerebellum, the basal ganglia, the hypothalamus, the hippocampus, the cerebral cortex, and the anterior cingulate cortex. Z-scores and *p*-values were calculated to measure the statistical significance of the predicted gene expression levels in each tissue type.

### 4.4. Functional Analysis

Functional pathway analysis was carried out in clusterProfiler v. 4.6.0 [49]. For gene set enrichment analysis (GSEA) in each tissue type, the algorithm was fed a list of genes from KEGG [50] (Release 105.0; 1 January 2023) and the Human Molecular Signatures Database (MSigDB) (H: hallmark gene sets) ranked by their *p*-values in ascending order. A set of functional pathways associated with cognitive impairment was generated for each brain area. The *p*-value adjusted for multiple testing was set at ≤0.05. The pathways were visualized using the R Data Visualization Package v. 4.2.2. 

## 5. Conclusions

In this study, we combined the advantages of a transcriptome-wide association study and a functional analysis to investigate the molecular and genetic mechanisms of cognitive impairment in people aged 90 and older. This approach allowed us to detect changes in gene expression levels in different brain tissues and assess the overall effect of these changes on cellular metabolism. An autophagy-related decrease in LRRC25 expression in the cerebral cortex may be associated with cognitive impairment. The results also indicate that there are associations between cognitive impairment and cell apoptosis, impaired autophagy, and oxidative stress. Together, these processes reflect brain aging and contribute to the higher susceptibility of older people to neurodegenerative diseases. The increased expression level of NADH oxidoreductase in the cerebral cortex of patients with cognitive impairment is of particular interest. The presented findings may facilitate future in silico and in vitro studies of cognitive impairment.

## 6. Limitations

The associations between functional signaling pathways and cognitive impairment were clinically significant. However, some limitations apply to the findings of this study. Since it is impossible to sample a living human brain, the conclusions are based solely on predicted changes in gene expression levels and genomic factors. The present study did not consider other factors that may affect gene expression levels.

## Figures and Tables

**Figure 1 ijms-25-03344-f001:**
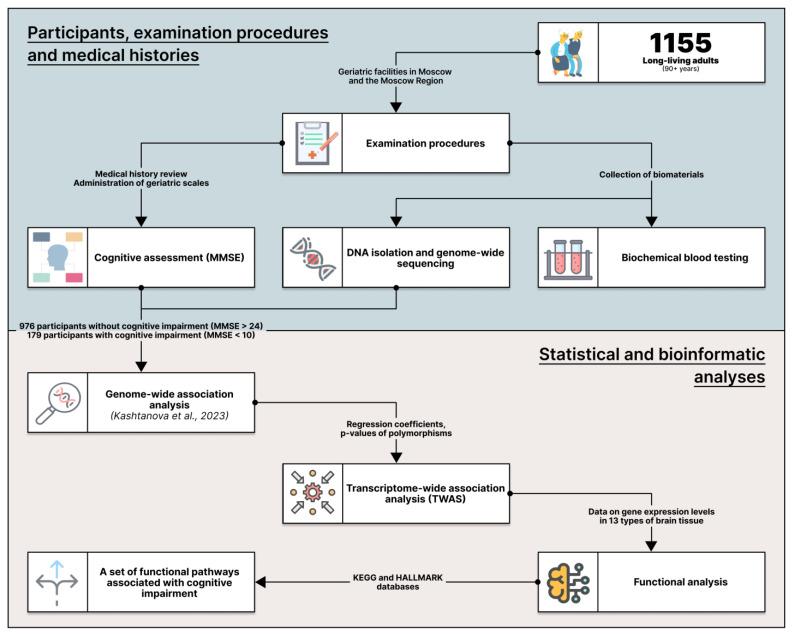
Scheme of the study design [5].

**Figure 2 ijms-25-03344-f002:**
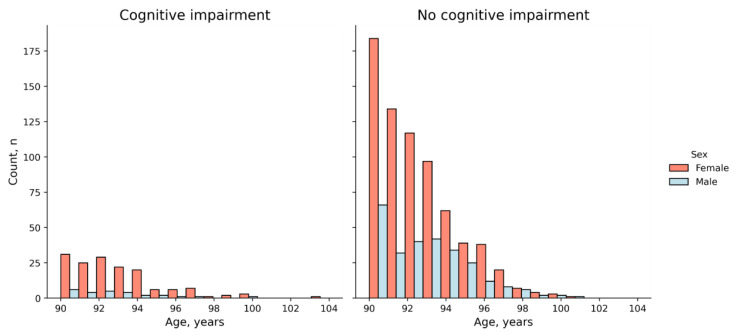
Age and sex composition of the study cohort.

**Figure 3 ijms-25-03344-f003:**
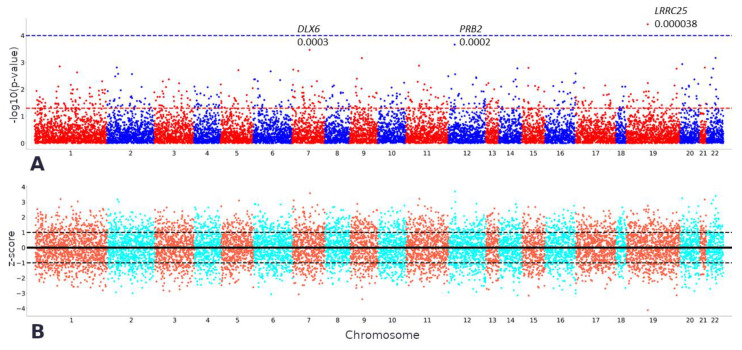
TWAS results for the anterior cingulate cortex. Names of genes with a *p*-value of less than 0.0003 are provided. (**A**). *p*-values of predicted gene expression levels. The red dotted line reflects the *p*-value level equal to 0.05, the blue line—0.0001. (**B**). Z-score of predicted gene expression levels.

**Figure 4 ijms-25-03344-f004:**
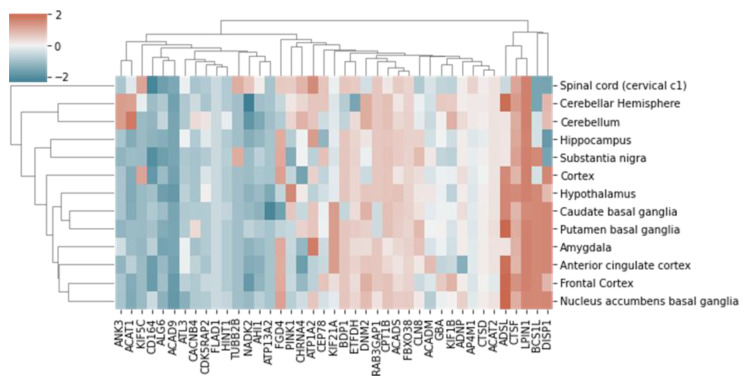
Predicted changes in expression levels of cognitive impairment-associated genes in different areas of the brain. The color scale reflects changes in expression levels (z-score) calculated using the TWAS results.

**Figure 5 ijms-25-03344-f005:**
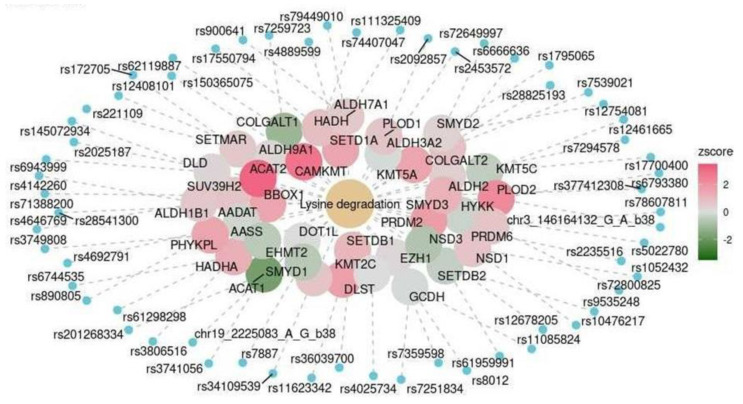
Section of the functional lysine degradation pathway (KEGG: hsa00310). The color scale reflects the predicted changes in gene expression levels in the cerebellum of participants with cognitive impairment.

**Figure 6 ijms-25-03344-f006:**
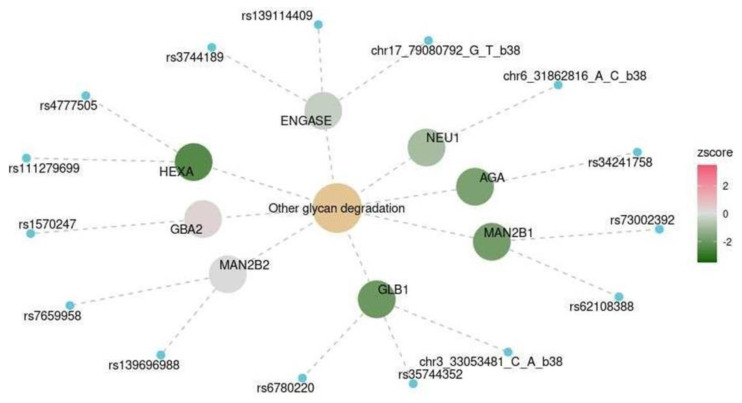
Functional glycan degradation pathway (KEGG: hsa00310). The color scale reflects the predicted changes in gene expression levels in the spinal cord at the level of C1 in participants with cognitive impairment.

**Figure 7 ijms-25-03344-f007:**
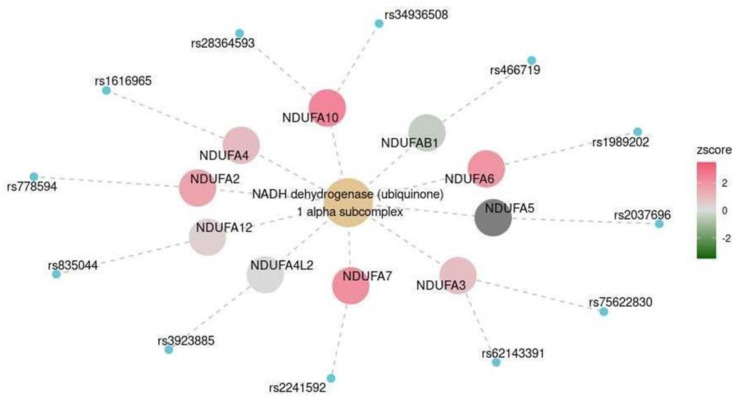
Functional pathway for the synthesis of the NADH: ubiquinone dehydrogenase in the cerebral cortex. The color scale reflects the predicted changes in gene expression levels in participants with cognitive impairment.

**Figure 8 ijms-25-03344-f008:**
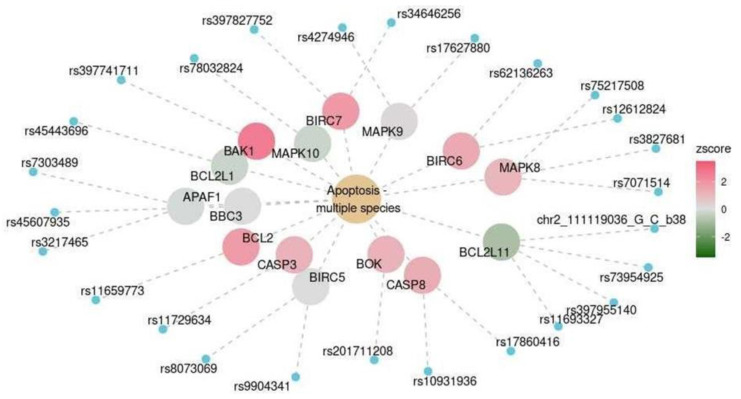
Functional pathway for apoptosis in the cerebellum. The color scale reflects the predicted changes in gene expression levels in participants with cognitive impairment.

**Table 1 ijms-25-03344-t001:** Changes in the incidence of cognitive impairment in response to a 1-unit increase in the level of serum biochemical markers.

Marker	OR (CI)	*p*-Value (Adjusted for Multiple Testing)
Glucose, µmol/L	0.90 (0.77; 1.06)	1
Insulin, μIU/mL	0.94 (0.91; 0.97)	0.008956
Apolipoprotein A, mg/dL	0.97 (0.96; 0.98)	1.91 × 10^−14^
Apolipoprotein B, mg/dL	0.996 (0.990; 1.003)	1
Total cholesterol, µmol/L	0.86 (0.74; 0.997)	0.73916
LDL, µmol/L	1.01 (0.85; 1.2)	1
HDL, µmol/L	0.22 (0.12; 0.37)	5.76 × 10^−7^
IGF-1, ng/mL	0.99 (0.985; 0.995)	0.000763
Cortisol, nmol/L	1.0003 (0.9995; 1.001)	1
25(OH) D, ng/mL	0.89 (0.86; 0.93)	7.74 × 10^−7^

Note: Gender, age, and BMI were used as covariates in the logistic regression model. OR: odds ratio; CI: confidence interval.

**Table 2 ijms-25-03344-t002:** Statistically significant results of the functional analysis and annotations from KEGG and HALLMARK.

Brain Region	Functional Pathway			*p*-Value (Adjusted for Multiple Testing)
	Database	Name	ID	
Spinal cord at the level of C1	KEGG	Other glycan degradation	hsa00511	0.041
Nucleus accumbens	HALLMARK	HALLMARK_P EROXISOME	HALLMARK_P EROXISOME	0.038
Cerebellum	KEGG	Apoptosis–multiple species	hsa04215	0.016
Cerebellum	KEGG	Lysine degradation	hsa00310	0.016
Cerebral cortex	MKEGG	NADH dehydrogenase (ubiquinone) 1 alpha subcomplex	M00146	0.047
Frontal cortex	MKEGG	NADH dehydrogenase (ubiquinone) 1 alpha subcomplex	M00146	0.003

## Data Availability

The original contributions presented in the study are included in the article/Appendix A, further inquiries can be directed to the corresponding author.

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
