# Peer review of "Cognitive Impairment in Nonagenarians: Potential Metabolic Mechanisms Revealed by the Synergy of In Silico Gene Expression Modeling and Pathway Enrichment Analysis"

_ijms, 2024, doi:10.3390/ijms25063344_

Round 1
Reviewer 1 Report
Comments and Suggestions for Authors
My suggestions:
1. Some genes in TWAS (ACE, TOMM40, etc) were also described as potential genetic risk factors for AD. Authors may mention this a little more in detail.
2. The authors only considered the participants in Moscow and nearby Moscow. It would be interesting in the future to expand this study to other regions of Russia too in the future (since it is a large country, and due to the possible geographical isolation, differences may be possible in gene expression patterns).
3. As I understood, the authors only considered elderly patients and healthy individuals (90 years or older). Is it possible that altered gene expression of these genes may be possible among younger patients with cognitive impairment?
4. In the Results section, I would add a table, that summarizes the most significant differently expressed genes, whether they are under-or overexpressed in patients, compared to controls, and their function in cognition.
5. Figure 3 may be uploaded in better resolution.
6. Is the disease-related variant in LRRC25 located in the coding region or is it an intronic variant?
Comments on the Quality of English Language
English is generally fine.
Author Response
Thank you ever so much for your insightful feedback and the opportunity to improve the manuscript. Below, we address each of your comments in detail.
1. Indeed, these genes are widely known to be involved in cognitive impairment. However, in our study, we did not find significant associations between cognitive impairment and the above genes. Therefore, a detailed discussion of these genes was deemed beyond the intended scope and focus of the current manuscript.
2. Thank you for your valuable suggestion. We could not agree more that expanding the study could yield valuable insights that further our research goals. Participant recruitment and sample collection efforts are now under way, and we expect to have finished this next stage of our research within the next few years.
3. Altered gene expression of these genes may be possible among younger patients. However, we cannot confirm or reject this hypothesis. Many biochemical, genetic, and epigenetic characteristics are unique to long-living adults (those 90 years of age or older), and these unique characteristics are rarely or never found in younger individuals. Moreover, cognitive impairment in older adults and younger individuals may have different pathogeneses. For instance, early-onset and late-onset Alzheimer’s disease (ca. 30 years vs. ca. >65 years) are known to have different genetic and environmental causes. Research focused specifically on a younger sample is find answers to this question.
4.The summary table of the most significant differently expressed genes is presented in the Supplementary materials as Table S1. Due to the size of the table, we opted not to include it in the main text. For convenience, a reference to Table S1 has been added to the results section (page 4, section 2.2., paragraph 1, line 119).
5. Thank you for pointing this out. Figure 3 will be uploaded to the submission system in better resolution.
6. In this study, we did not focus on individual variants but rather on changes in gene expression levels caused by all variants, both exonic and intronic, and the effects of these changes on cognitive status.
Reviewer 2 Report
Comments and Suggestions for Authors
1.How did the study address the limitations of previous research on cognitive impairment, specifically in long-living adults?
2. How did the study integrate the results of the genome-wide association study with the PrediXcan gene prediction algorithm?
3. What was the criteria for sample collection? How did the authors arrive at specific cognitive scoring?
4. Cognitive impairment was encoded as 1 and cognitive health was encoded as 0. The Authors can add background reference.
5. What was the rationale behind selecting blood. Why not serum or plasma?
6. Why didnt the authors choose to perform PPI network or clustering approach for their interactive behaviour?
7. "An Updated Review: Androgens and Cognitive Impairment in Older Men". How did the authors arrive at this statement -"men may be less susceptible to age-related cognitive impairment". Please refer that article and compare your findings.
8. The authors didnt conclude clearly. Can present their identified genes and their pathways better.
Author Response
Thank you ever so much for your insightful feedback and the opportunity to improve the manuscript. Below, we address each of your comments in detail.
1. Transcriptome-wide association studies help measure gene expression changes specifically in the brain. Thus, they help address the limitations inherent in live brain studies.
2. As mentioned in the Methods section, we used the results of the genome-wide association study, such as detected SNVs, logistic regression coefficients, and p-values, as the input for the PrediXcan algorithm. Therefore, PrediXcan predicted gene expression changes caused by a set of SNVs that we found to be associated with cognitive impairment.
3. Our recruitment procedure was based on an “all-comers” design, with age being the only inclusion criteria: all participants had to be ≥90 years old. In the Methods section (subsection 4.1, line 327), we cite our previous study, which describes the sample collection procedure in detail: Kashtanova DA, Taraskina AN, Erema VV, Akopyan AA, Ivanov MV, Strazhesko ID, et al. Analyzing Successful Aging and Longevity: Risk Factors and Health Promoters in 2020 Older Adults. Int J Environ Res Public Health. 2022;19. doi:10.3390/ijerph19138178.
To find the most significant differences between cognitively impaired and cognitively healthy participants, we compared those with MMSE scores of ≤ 9 (Group 1) and those with MMSE scores of > 24 (Group 0).
4. Thank you for the recommendation. This information has been added to the Methods section (subsection 4.1; lines 329–330).
5. Whole blood was used for DNA extraction. Serum was used to obtain data on the concentration of biochemical markers (Table 1). We have added this clarification to the table title.
6. Thank you for your suggestion. We will most definitely try out this method in our future studies. However, in this study, we focused on the GSEA analysis of TWAS results. GSEA can also highlight interactive behavior of genes, the expression of which significantly differs in cognitively impaired and cognitively healthy people.
7. Thank you for pointing this out. However, the work you cited discusses the reasons for cognitive decline in older men compared to younger ones. The statement “men may be less susceptible to age-related cognitive impairment” in our work suggests that men aged 90+ years are less susceptible to cognitive impairment than women at the same age. However, the article “An Updated Review: Androgens and Cognitive Impairment in Older Men” suggests an interesting metabolic pathway in which androgens act as inhibitors of oxidative stress. This, in turn, may be one of the reasons why men have an additional protective factor compared to women. We have added this statement to the discussion section.
8. Thank you for your suggestion. Changes have been made to the Conclusion section.
Round 2
Reviewer 1 Report
Comments and Suggestions for Authors
The authors fulfilled my suggestions. Thank you.